# Toward Self-Improvement of LLMs via Imagination, Searching, and Criticizing

**Ye Tian[1,2][*], Baolin Peng[1][*], Linfeng Song[1][*], Lifeng Jin[1], Dian Yu[1], Lei Han[2]**
**Haitao Mi[1][†], Dong Yu[1]**
[1]Tencent AI Lab, Bellevue, WA
[2]Tencent Robotics X
{baolinpeng,lfsong,lifengjin,yudian,haitaomi,dyu}@global.tencent.com
{yaptian,lxhan}@tencent.com

## Abstract

Despite the impressive capabilities of Large Language Models (LLMs) on various tasks, they still struggle with scenarios that involves complex reasoning and planning. Self-correction and self-learning emerge as viable solutions, employing strategies that allow LLMs to refine their outputs and learn from self-assessed rewards. Yet, the efficacy of LLMs in self-refining its response, particularly in complex reasoning and planning task, remains dubious. In this paper, we introduce ALPHALLM for the self-improvements of LLMs, which integrates Monte Carlo Tree Search (MCTS) with LLMs to establish a self-improving loop, thereby enhancing the capabilities of LLMs without additional annotations. Drawing inspiration from the success of AlphaGo, ALPHALLM addresses the unique challenges of combining MCTS with LLM for self-improvement, including data scarcity, the vastness search spaces of language tasks, and the subjective nature of feedback in language tasks. ALPHALLM is comprised of prompt synthesis component, an efficient MCTS approach tailored for language tasks, and a trio of critic models for precise feedback. Our experimental results in mathematical reasoning tasks demonstrate that ALPHALLM significantly enhances the performance of LLMs without additional annotations, showing the potential for self-improvement in LLMs. The code is available at `https://github.com/YeTianJHU/AlphaLLM`.

## 1 Introduction

LLMs, trained on trillions of tokens with billions of parameters have shown unparalleled capabilities in a wide range of natural language processing tasks (Touvron et al., 2023b; Team et al., 2023; OpenAI, 2023). Nevertheless, they continue to face challenges in scenarios requiring complex reasoning and strategic planning (Valmeekam et al., 2022; Stechly et al., 2024). While advanced prompting approaches such as Chain, Tree, Graph-of-Thought (Wei et al., 2022; Yao et al., 2024; Besta et al., 2024; Ding et al., 2023), it remains essential to fine-tune LLMs using a substantial volume of high-quality, supervised data to fundamentally improve the model performance (Nye et al., 2021; Lewkowycz et al., 2022; Chung et al., 2022). This methodology is inherently limited by the scope and quality of data that humans can provide.

Considering these challenges, the concept of self-correction and self-learning have been proposed as promising solutions (Madaan et al., 2024; Saunders et al., 2022; Chen et al., 2024). Within these framework, LLMs typically operate by employing two main strategies: 1) they continuously refine

---

[*]Equal Contribution; [†]Corresponding Author

38th Conference on Neural Information Processing Systems (NeurIPS 2024).

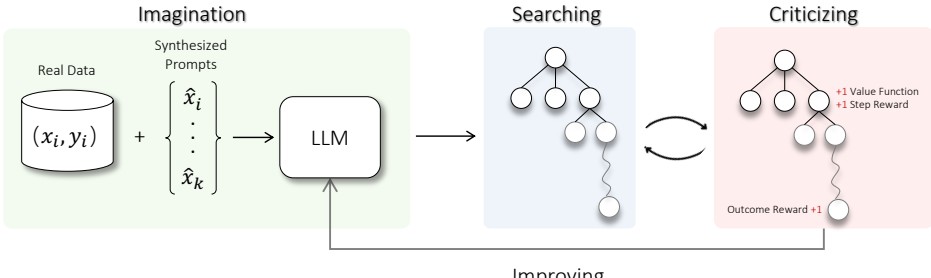

Figure 1: Imagination-Searching-Criticizing self-improvement loop: Imagination component synthesizes prompts as new learning examples, with MCTS searching better trajectories guided by signals from critics for policy improving.

their responses based on the feedback of their past responses, and 2) they extensively sample responses then learn from preferences judged by itself as reward models with PPO or DPO (Yuan et al., 2024a,b; Chen et al., 2024). However, it remains a matter of ongoing research whether LLMs can effectively critique their own outputs to either enhance response quality or apply a scalar reward to indicate the quality of responses, especially in contexts demanding intricate planning and reasoning (Valmeekam et al., 2022; Stechly et al., 2024; Huang et al., 2023; Hong et al., 2023). On the other hand, advanced search algorithms such as MCTS, combined with reinforcement learning, have enabled models to learn from self-play and achieve human parity or even surpass human performance in complex tasks such as the game of Go (Silver et al., 2016, 2017). This naturally raises a question: is it viable to leverage the strengths of MCTS alongside LLMs to inaugurate a novel paradigm of self-improving? More precisely, could the assimilation of MCTS empower LLMs to more effectively explore better responses, guided by strategic signals, and subsequently optimize these responses to enhance overall performance?

To answer this question, we begin with a systematic examination of AlphaGo, identifying three critical aspects for its success: (*i*) The large volume of data, including self-play data. (*ii*) The use of tree search, which facilitates the exploration of potential moves through statistical sampling of the large search space. (*iii*) Accurate and unambiguous environment feedback; the direct and accurate feedback (win or loss) provided by the game of Go offers a clear and unequivocal learning signal (Silver et al., 2017). The integration of MCTS with LLMs for self-improvement has several challenges: (*i*) Limited Data: High-quality annotated data for LLMs is generally scarce. Furthermore, how to construct of synthetic data for LLMs training, similar to AlphaGo's self-play data, remains unclear. (*ii*) Search Efficiency: The vast number of potential token combinations in natural language tasks results in an exponentially large search space, posing a significant challenge to the efficiency of MCTS (Ramamurthy et al., 2022). (*iii*) Imperfect Feedback: In contrast to the clear win/loss feedback in Go, feedback in natural language tasks is often subjective and nuanced, without a straightforward measure of success.

In this paper, we introduce ALPHALLM, an imagination-searching-criticizing framework designed for the self-improvement of LLMs . ALPHALLM consists of three key components, as illustrated in Figure 1. First, an imagination component is designed to synthesize prompts, alleviating the issues of data scarcity. Second, we propose $\eta$MCTS tailored for efficient searching in language tasks. Particularly, it has been show that planning at multiple levels of temporal abstraction is critical for RL problems with a long horizon and large action space (Sutton et al., 1999b; Peng et al., 2017; Luketina et al., 2019). As such, we propose formulating the text generation process as options over a Markov Decision Process (MDP) problem, where each option represents the generation of a collection of tokens for a specific subtask, similar to the concept of chains in chain-of-thought prompting. This formulation improves search efficiency by substantially reducing the search depth. Additionally, we propose the use of state merge and adaptive branching factors to further enhance search efficiency by balancing the trade-off between search width and depth. Lastly, since accurate feedback is crucial to the success of MCTS, we introduce a trio of critic models to guide $\eta$MCTS, including a value function for estimating expected rewards, a process reward model for assessing node correctness, and an outcome reward model for evaluating the overall trajectory. For complex tasks with which LLMs struggle assessing such as arithmetic computation and code execution, to ensure the accuracy

of feedback, we augment the critics with the capacity to make dynamic decisions on which tools to use, when to use them, and how to use them effectively. After $\eta$MCTS stage, we collect the trajectory with the largest reward from the critic models as the training examples to improve LLMs.

The experimental results on mathematical reasoning tasks demonstrate that ALPHALLM can efficiently search for better responses and use them to improve LLMs' performance, forming an effective self-improving loop. Notably, based on Llama-2-70b and WizardMath-70B-V1.0, ALPHALLM can improve its performance from 57.8 to 92.0 on GSM8K and from 20.7 to 51.0 on MATH, performing comparably to GPT-4.

## 2 Related Work

**Search with LLM**   Effective search strategy has been shown crucial for tasks that involve complex reasoning and planning, such as go (Silver et al., 2016) and math reasoning (Cobbe et al., 2021; Hendrycks et al., 2021). For math reasoning tasks, various search methods have been studied. One direction of research (Zhu et al., 2024; Xie et al., 2024) designed beam search with dynamic pruning, where beam items of low quality are pruned. Another line of work (Yao et al., 2024; Long, 2023; Besta et al., 2024; Hao et al., 2023; Feng et al., 2023) maintains a tree or a graph that represents the current progress of solving the input question where potential branches are iteratively expanded. Both our approach and Feng et al. (2023) are based on the MCTS algorithm, while one main difference is how to define a search step: Feng et al. (2023) fix a search step to be either a token or a sentence, while our approach is more flexible on deciding steps. We have also carefully designed the MCTS process, incorporating multiple critique signals to guide the search more effectively and introducing adaptive search parameters for improved state exploration. As the result, our approach achieves much better performances.

**LLM Self-improving**   Being a key to the success of scalable oversight (Bowman et al., 2022), self-improving for LLM aims to align the LLM to human preference and values mainly using the supervision from the knowledge inside the LLM (Zelikman et al., 2022, 2024). One crucial part of self-improving is how to obtain reliable signal of critique to distinguish between good responses from the LLM and bad ones. Initial work (Bai et al., 2022; Wang et al., 2022) first asks the LLM to generate input queries of diverse tasks and the corresponding outputs. They then rely on hand-crafted heuristic rules to filter out redundant or low-quality data pairs (e.g. the query is too long or too short). Since it is non-trivial to compose effective heuristic rule, later work (Sun et al., 2023; Li et al., 2023; Guo et al., 2024) proposes a few general principles or judging criteria and ask the LLM itself to evaluate the quality its responses based on these guidance, hoping that LLMs can automatically designate these principles into each data point to better guide data filtering. However, this requires LLMs to have strong abilities to apply these principles for each specific case and make correct judgements. Different from previous work, we propose to leverage the supervision from MCTS for LLM self-improvement: taking the outputs of MCTS to continue train the LLM. This is because the outputs from MCTS are usually in much better quality then standard nucleus sampling, and the large gap ensure that the LLM can self improve.

## 3 Preliminaries

### 3.1 Problem Formulation

In this paper, we consider a LLM characterized by probability $p_\theta$ and denoted as policy $\pi_\theta$. It takes a sequence $\boldsymbol{x} = [x_1, \cdots, x_n]$ as input, which is typically referred as prompt, to generate the response $\boldsymbol{y} = [y_1, \cdots, y_m]$. In the context of LLMs, each $x_i$ and $y_i$ represents a token from a pre-defined vocabulary. The policy $\pi_\theta$ operates in an autoregressive manner, where each token is generated sequentially, relying solely on the context provided by the previously generated tokens. The policy therefore constitutes a Markov process in which the conditional probability distribution $p_\theta(\boldsymbol{y}|\boldsymbol{x})$ can be decomposed and expressed with the chain rule as $p_\theta(\boldsymbol{y}|\boldsymbol{x}) = \prod_{i=1}^{m} p_\theta(y_i|\boldsymbol{x}, \boldsymbol{y}_{<i})$.

With this property, the text generation task can be formulated as an Markov Decision Process (MDP) problem consisting of $(\mathcal{S}, \mathcal{A}, T, R, \gamma)$ in which, $\boldsymbol{s}_t \in \mathcal{S}$ represents the context information of current trajectory, *i.e.,* current status of the generation process, *e.g.,* a partial response to a prompt; $a_t \in \mathcal{A}$ denotes a single action or sampled token from the vocabulary, leading to a transition to a new state

$s_{t+1}$, by concatenating $s_t$ and $a_t$; $r_t = R(s_t, a_t)$ manifest the evaluation of the generation to the prompt, reflecting the desirability or preferences of each state-action pair.

This MDP framework sets the stage for applying Reinforcement Learning (RL) methods to optimize the policy $\pi_\theta$ aiming to maximize the expected cumulative reward $R$. Base on these setups, we describe the self-improving problem. Given a LLM $\pi_\theta$ and an initial dataset $\mathcal{D}^0$, which consists of $N$ expert-generated prompt-response pairs $\{(x_i^0, y_i^0) \mid i \in [N]\}$, the goal of self-improving is to iteratively refine $\pi_\theta$ to maximize the reward. The refinement process includes learning from synthesized prompts and corresponding responses. These responses are obtained using an advanced search algorithm that navigates the space of possible responses to maximize the expected reward. The detailed process is described in Algorithm 1 in Appendix. The primary challenges in forming an effective self-improving loop lie in synthesizing suitable prompts, efficiently searching over a vast action space, and obtaining precise feedback, which will be discussed in §4.

## 3.2 Monte Carlo Tree Search

MCTS is a sampling-based search algorithm for policy optimization in decision-making problems. It would iteratively build a search tree, by repeating four phases: selection, expansion, evaluation, and backpropagation. In the selection phase, it would recursively select the children from the root node by Upper Confidence Bound (UCB) (Auer et al., 2002), $UCB(i) = w_i + C * \sqrt{2 * \ln \frac{N_i}{n_i}}$, where $n_i$ and $N_i$ are the visit counts for the node $i$ and its parent respectively, $C$ represents a hyperparameter balancing exploration and exploitation, and the $w_i$ is the average value of all descendant nodes of $i$.

# 4 ALPHALLM

## 4.1 Overview

The architecture of ALPHALLM is depicted in Figure 1, comprising three key components. Firstly, the imagination component is tasked with synthesizing prompts as learning examples. Secondly, an efficient search component, named $\eta$MCTS, is proposed to search high-quality trajectories for optimizing the policy. Lastly, the search process is guided by critics specifically designed to provide reliable signals.

## 4.2 Data Synthesizing

Let $\mathcal{D}^0 = \{(x_i, y_i) \mid i \in [N]\}$ denote the initial dataset consisting of $N$ expert-generated prompt-response pairs. The data synthesizing process aims to expand this dataset by generating a set of synthesized prompts $\mathcal{D}^1 = \{(x_i^1, \cdots) \mid i \in [N]\}$. The generation of each synthesized prompt $x_i^1$ can be mathematically described as a transformation $g$ applied to one or more examples from $\mathcal{D}^0$, $x_i^1 = g(x_{i_1}^0, \cdots, x_{i_m}^0, \pi^0)$ where $x_{i_1}^0, \cdots, x_{i_m}^0$ are selected examples from $\mathcal{D}^0$. The transformation function $g$ controls the synthesis process, which can be a learnable function, manually defined heuristic rules, a strong LLM or the policy model itself $\pi^0$ equipped with data synthesis instructions. The data synthesizing process aims to enrich the diversity and complexity presented for the training of the policy model. Among various strategies, such as Self-instruct (Wang et al., 2022), Evol-instruct (Xu et al., 2023), we opt for a method akin to that described in Yu et al. (2023).

## 4.3 $\eta$MCTS

### 4.3.1 Option-level MCTS

When applying MCTS to LLMs, it is natural to perform token-level search, where each token is considered as an action (Liu et al., 2023). However, the substantial vocabulary size typical of LLMs presents a significant challenge *i.e.,* conducting a deep search in such a vast space becomes increasingly complex as the search space expands exponentially. To mitigate this, some efforts proposed a sentence-level search, treating each sentence or step as a search node (Feng et al., 2023). While this method reduces the search space, it might compromise the flexibility and effectiveness of applying MCTS to LLMs, which is particularly true for tasks where subtle variations in token can dramatically impact the outcome, or where a more comprehensive search beyond a sentence is necessary.

| Search Node | Example | Termination |
|---|---|---|
| Token-level | $y_0 \to y_1 \to y_2 \to y_3 \to y_5 \to y_6 \to y_7 \to y_8$ | token |
| Sentence-level | $y_0 y_1 y_2 \boxed{\hookleftarrow} \to y_4 y_5 y_6 \boxed{\hookleftarrow} \to y_7 y_8 y_9 y_{10}$ | new line |
| Option-level | $y_0 \to y_1 y_2 \boxed{\hookleftarrow} \to y_4 y_5 y_6 \boxed{\hookleftarrow} y_7 y_8 y_9 \boxed{\hookleftarrow} \to y_{10}$ | termination function |

Table 1: Comparative illustration of token-level, sentence-level, and option-level MCTS search nodes. $y$ denotes a token sampled from the policy model. The arrow $\to$ represents the transition from one search node to the subsequent node within the search process.

Inspired by Sutton et al. (1999a); De Waard et al. (2016), we use the term option as a search node and propose option-level MCTS where each option represents a sequence of tokens, which can range from multiple tokens to several sentences. A comparisons of different levels search is listed in Table 1. Mathematically, an option $o = \langle \mathcal{I}, \pi, \beta \rangle$, where $\mathcal{I} \subseteq \mathcal{S}$ is a set of initial states for the option; $\pi : \mathcal{S} \times \mathcal{A} \to [0, 1]$ is a policy to generate actions, which in our case is a LLM; and $\beta : \mathcal{S}^+ \to [0, 1]$ is the termination function. Starting from a state $s_t$, we can choose all the options for which $s_t \in \mathcal{I}$. Once an option is chosen, the policy $\pi$ will generate actions for several steps until the option terminates according to the termination function $\beta$. The option-level MCTS consists of stages including selection, expansion, simulation, and backpropagation. The option-level formulation offers more flexibility compared to the sentence-level, as a new line can be treated as a special case of the termination function, as demonstrated in Table 1. Additional detailed steps of the option-level MCTS can be found in Appendix A.2.

### 4.3.2 Importance-Based Adaptive Branching

In previous works related to option/sentence level tree search (Feng et al., 2023; Yao et al., 2024), it was a common practice to assume that each node in the tree has the same predefined width, *i.e.*, branching factor. This assumption was due to the fact that unlike token-level MCTS with a limited action space, the sample space at the option-level is exceedingly large, with an unlimited number of token combinations. As a result, it was necessary to set a predefined maximum width for each node. However, this predefined branching factor is hard to set, as an improper choice can lead to a search tree that is either too shallow or too thin, resulting in an inefficient exploration of the search space.

To quantify the error induced by the branching factor limit, we defined the branching error $E_\phi(t)$. For a node $t$ with a branching factor of $m_t$, it aims to use the $m_t$ child options $\boldsymbol{o}_t^i \sim \mathcal{D}_t^{children}$ (where $i \in \{1, \ldots, m_t\}$) to represent all possible options. Consequently, for a legal option $\boldsymbol{o}_t^j \sim \pi(\boldsymbol{s}_t)$ from the option space, we can calculate the minimal value difference between it and the $m_t$ existing options, which captures the error associated with representing other possible options using the $m_t$ available options. It can be formulated as $E_\phi(t) = \mathbb{E}_{\boldsymbol{o}_t^j \sim \pi(\boldsymbol{s}_t)}[\min_{\boldsymbol{o}_t^i} |v_\phi^\pi([\boldsymbol{s}_t, \boldsymbol{o}_t^j]) - v_\phi^\pi([\boldsymbol{s}_t, \boldsymbol{o}_t^i])|]$, where $v_\phi^\pi$ is the value function which will be detailed in §4.4. Here we define the importance of node $\boldsymbol{s}_t$ as $I(\boldsymbol{s}_t) = \max_{\boldsymbol{o}_t^i} |v_\phi^\pi([\boldsymbol{s}_t, \boldsymbol{o}_t^i]) - v_\phi^\pi(\boldsymbol{s}_t)|$. For simplicity, we assume that the value of the children nodes are uniformly distributed (a detailed analysis of the Gaussian distribution can be found in Appendix A.4). Under this assumption, we show in Appendix A.3 that $E_\phi(t) \leq \frac{I(\boldsymbol{s}_t)}{m_t - 1}$. While $E_\phi$ is less than some $\epsilon$, we aim to use a smaller total number of nodes for efficiency.

**Theorem 4.1.** *The optimal branching factor $m_t$ in a tree search is set such that $m_t - 1$ is proportional to the node importance $I(\boldsymbol{s}_t)$, under the condition $\frac{I(\boldsymbol{s}_t)}{m_t - 1} \leq \epsilon$. Refer to Appendix A.3 for the detailed proof.*

A similar concept has also been proposed in Taylor et al. (2014); Clouse (1996). Intuitively, $I(\boldsymbol{s}_t)$ captures the maximum value deviation from the current state. When this value is small, there is no need to explore further on this node, as there will not be a significant difference by rolling out on this node. Conversely, if the value is large, it is worth trying different children. We set the number of children allowed for a node $n(\boldsymbol{s}_t)$ (after extracting 1) to be linear with this importance, using a factor $\alpha$. In practice, to avoid extreme cases of large variance of $I(\boldsymbol{s}_t)$ in the early stage, we bound the number of children by depth-dependent constants $c_{\min}(t)$ and $c_{\max}(t)$, $n(\boldsymbol{s}_t) = \max\left(c_{\min}(t), \min\left(\lfloor \alpha I(\boldsymbol{s}_t) \rfloor + 1, c_{\max}(t)\right)\right).$

### 4.3.3 State Merge

With $n(\boldsymbol{s}_t)$ determined, another issue is that options under the same node may be very similar, leading to many unnecessary sub-trees. Since we cannot directly control the $\boldsymbol{o}_t \sim \pi(\boldsymbol{s}_t)$, one strategy to mitigate this issue is to utilize the concept of move groups, as discussed in Van Eyck & Müller (2012). By merging similar nodes into the same group, we can increase the diversity among groups, thereby covering a larger problem space with limited search rollouts and making the search process more efficient.

Here, we adapt the definition of node predicate $p_{vM}$ from Abel et al. (2018) and Fu et al. (2024) to represent whether two nodes are extremely similar. In practice, each time we generate a new option from the policy, we use heuristic functions as $p_{vM}$ to check its similarity with all existing groups. The heuristic function can either be a faster rule-based measurement (e.g., edit distance) or a model-based method (e.g., prompting a language model). Based on this, we decide whether to merge this option with a previous one or create a new group.

### 4.3.4 Fast Rollout with Specialized LM

The simulation operation which employs a rollout policy to project future trajectories from a given state, is crucial for an effective MCTS. This process significantly improves the efficiency of exploration and exploitation, and enhances the accuracy of reward estimation[2]. Estimations made at the end of trajectories tend to have lower bias but higher variance; thus, simulating multiple possible trajectories yields low-bias, low-variance estimates, enabling a more informed and effective search process. Ideally, $\pi_\theta$ would serve as the rollout policy, yet its computational demands render it impractical for the rapid simulations required by MCTS. To address this challenge, we propose the use of a smaller, specialized LM as the fast rollout policy $\pi^{\texttt{fast}}$. Given a state $\boldsymbol{s}_t$, the fast rollout policy $\pi^{\texttt{fast}}$ efficiently continues generation until it reaches a termination condition, denoted as $\pi^{\texttt{fast}}(\boldsymbol{s}_t)$.

### 4.4 Critic

In ALPHALLM, we design three types of critic models to guide the search process.

**Value Function** The value function, denoted as $v^\pi(\boldsymbol{s})$, represents the expected return starting from state $\boldsymbol{s}$ and following policy $\pi$ thereafter, given by $v^\pi(\boldsymbol{s}) = \mathbb{E}_{\tau \sim \pi}[R(\tau)|s_0 = \boldsymbol{s}]$ where $R(\tau)$ represents the discounted return of trajectory $\tau$. To train a parameterized value function $v^\pi_\phi(\boldsymbol{s})$, given the prompts $\mathcal{D} = \{(\boldsymbol{x}_i, \cdots) \mid i \in [N]\}$, for each prompt $\boldsymbol{x}_i$, we generate multiple trajectories $\boldsymbol{\tau}_i^j = \{\boldsymbol{x}_i, \boldsymbol{o}_{i1}^j, \boldsymbol{o}_{i2}^j, \cdots, \boldsymbol{o}_{iT}^j\}$ by following policy $\pi$ for $J$ times. A final reward $r_i^j$ is assigned to indicate whether $\boldsymbol{\tau}_i^j$ aligns with $\boldsymbol{y}_i$—for example, rewarding trajectories that contain correct answers in mathematical tasks or closely follow instructions as ground truth. We then construct a dataset $\mathcal{D}_{\texttt{value}} = \{(\boldsymbol{s}_{it}^j, v_{it}^j) \mid i \in [N], t \in [T], j \in [J]\}$ where $\boldsymbol{s}_{it}^j = [\boldsymbol{x}_i \cdot \boldsymbol{o}_{<it}^j]$ and $v_{it}^j = r_i^j$. The value function $v^\pi_\phi$ is optimized by minimizing the mean squared error: $\mathcal{L}_\phi = -\mathbb{E}_{(\boldsymbol{s},v) \sim \mathcal{D}_{\texttt{value}}}(v^\pi_\phi(\boldsymbol{s}) - v)^2$. Similar to (Feng et al., 2023), $v^\pi_\phi$ is a LLM with an MLP layer on top to output a scalar on each token, using the scalar prediction at the last token of each state as the value.

**PRM** The value function often struggles with credit assignment problem (Sutton, 1984) and its learning could be inefficient due to delayed and sparse rewards (Sutton & Barto, 2018). Therefore, we propose to incorporate PRM that introduces process supervision (Lightman et al., 2023) for direct option assessment. PRM generates intrinsic rewards (Chentanez et al., 2004) to encourage explorations of advantageous options, effectively mitigating issues of reward sparsity by providing immediate, action-specific rewards. Given a state $\boldsymbol{s}_t$ and an option $\boldsymbol{o}_t$ at time $t$, the PRM aims to predict the immediate reward $r_t^{\texttt{PRM}}$ that results from taking option $\boldsymbol{o}_t$ in state $\boldsymbol{s}_t$. Formally, the PRM is a function $R(\boldsymbol{s}_t, \boldsymbol{o}_t) \to r_t^{\texttt{PRM}}$. While PRM ideally requires quality labels for each state (Uesato et al., 2022), due to the high cost and time involved in obtaining these, MC estimation with prefix sampling (Wang et al., 2023) is used as a proxy, which aligns with the objective of the value function. Instead of adding a MLP layer on top of the policy model for outputting a scalar reward (Ouyang et al., 2022), we formulate PRM as a text generation task to best leverage LLM's intrinsic knowledge

---

[2]Typically, the closer the simulation is to the termination state, the more accurate the reward estimation becomes.

for assessing the quality of an option. We adapt the dataset constructed for the value function as $\mathcal{D}_{\text{PRM}} = \{(s_{it}, o_t, r_t^{\text{PRM}}) | i \in [N], t \in [T]\}$ where $r_t^{\text{PRM}}$ is the textual description of the reward, *e.g.,* an option can be regarded as good if $v_{it}$ is larger than certain threshold. To train PRM, we initialize it from the policy model $\pi$ and use the following prompt templates and typical language model loss. The prompt template is shown in Appendix A.5.

**ORM**    In additional to the value function and PRM, ORM is also used to guide MCTS. ORM is designed to evaluate options sequences in their entirety, assessing the extent to which the complete trajectory aligns with the desired end goal (Uesato et al., 2022; Lightman et al., 2023; Wang et al., 2023; Feng et al., 2023). The outcome evaluation complements value function and PRM by offering a comprehensive assessment of trajectories. Crucially, ORM plays a vital role in the simulation stage of MCTS by providing more accurate signals on the terminal state, which in turn facilitates a more balance between exploration and exploitation strategies. ORM is formulated as a text generation task, similar to PRM. We leverage the same dataset for the value function training and construct $\mathcal{D}_{\text{ORM}} = \{(x_i, o_{1:T}^i, r_i^{\text{ORM}}) | i \in [N]\}$, where each instance includes a initial state or prompt $x_i$, a sequence of actions or options $o_{1:T}^i$ taken from that state, and a textual reward $r_i^{\text{ORM}}$ indicating the sequence's success or quality. Similarly, ORM is initialized from the policy model $\pi$ and the following prompt templates and language model loss are used for training. The prompt template is shown in Appendix A.5.

The final score evaluation of a state $s$ is a weighted sum of the value function, PRM, and ORM: $s(s) = \beta_{\text{value}} \cdot v_\phi^\pi(s) + \beta_{\text{PRM}} \cdot \text{PRM}(s) + \beta_{\text{ORM}} \cdot \mathbb{E}_{\tau \sim \pi^{\text{fast}}(s)}[\text{ORM}(\tau)]$, where $\tau \sim \pi^{\text{fast}}(s)$ represents trajectories starting from $s$ under $\pi^{\text{fast}}$, and $\beta_{\text{value}}, \beta_{\text{PRM}}, \beta_{\text{ORM}}$ are hyperparameters. In practice, we found that the value function model has better precision and calibration, while PRM has superior recall (Appendix A.10). Although ORM with fast rollouts provides low-bias, low-variance estimates, it still inherits some bias from $\pi^{\text{fast}}$. Thus, combining these critics yields a stronger evaluation signal.

### 4.5    Policy Self-Improvement

The policy improvement an iterative process with each iteration containing two main steps: *data generation* and *policy finetuning*.

**Data generation**    In this step, we assume to have the current policy $\pi_{\theta_k}$ and synthetic prompts $\mathcal{D}_k = \{x_1^k, \dots\}$ at the $k$-th round, where each $x_1^k$ represents a question. We obtain the corresponding training data $\mathcal{D}_k$ for policy $\pi_{\theta_k}$ by firstly performing $\eta$MCTS on $\mathcal{D}_k$ (§4.3) and then sampling a trajectory $y_i^k$ from the corresponding tree for each question $x_i^k$. Here we choose the trajectory that yield the highest critic score on the leaf node for each input question. Next, we filter out instances where the corresponding trajectory is substandard forming $\mathcal{D}_k = \{(x_i^k, y_i^k) \mid f(x_i^k, y_i^k) > \gamma\}$ where $f$ represents a function for quality scoring, and $\gamma$ indicates a threshold. There can be several ways to implement the function, and here we simply use the ORM (§4.4).

**Policy finetuning**    With the obtained training data $\mathcal{D}_k$, we organize the data into the prompt templates shown in Appendix A.5. Then the policy $\pi_{\theta_k}$ is finetuned using target-loss: $\mathcal{L}_{\theta_k} = \mathbb{E}_{(x_i^k, y_i^k) \sim \mathcal{D}_k} [\log \pi_{\theta_k}(y_i^k | x_i^k)]$, resulting in an updated policy $\pi_{\theta_{k+1}}$. We leave other training methods, such as DPO (Rafailov et al., 2023) or PPO (Schulman et al., 2017) in future work.

## 5    Experiments

### 5.1    Experiment Setups

ALPHALLM is generally applicable to a wide spectrum tasks. As an early exploration, in this paper, we conduct experiments on mathematical reasoning problems where the learning signals are clear to define *i.e.,* , final answer is correct or wrong. We choose to evaluate on two widely used datasets GSM8K (Cobbe et al., 2021) and MATH (Hendrycks et al., 2021). For GSM8K, we utilize the whole test set while for MATH, due to computation constraints, we utilize a subset following the same procedure of Lightman et al. (2023). We evaluate the performance of predicting answers correctly for policy models. In addition, we calculate the average rollouts, represented by the number of nodes in

the tree, as a measure of computational efficiency. We compare the performance of AʟᴘʜᴀLLM with a suite of proprietary model, including OpenAI's GPT-4 and GPT-3.5, Anthropic's Claude-2, as well as Google's PaLM-2 and the gemini model family. To ensure a fair and consistent evaluation, we employ CoT as our primary prompting method. Additionally, we conduct comparisons with strong open-source models, including Llama-2-70b (Touvron et al., 2023a) and WizardMath-70B-V1.0 (Luo et al., 2023).

We select Llama-2-70b as the policy model for the GSM8K dataset and WizardMath-70B-V1.0 for the MATH dataset. To construct the training dataset for the value function, `PRM` and `ORM`, we generate 50 trajectories for each prompt and construct the training target following Section 4.4. Both `PRM` and `ORM` are initialized using the weights from the policy model, while the value function uses a smaller Llama-2-13b model, as we observed no performance gains from increasing the value function model size. In the design of `ORM`, tool usage is not incorporated for GSM8K. However, for MATH, we enhance `ORM` by incorporating tools like python sympy to assess the quality of a trajectory, in a manner similar to that described by Gou et al. (2023). The training employ a learning rate of 1e-6 and are trained for one epoch. For the fast rollout policy model, we opt for the Abel-002-7B model (Chern et al., 2023) for both the GSM8K and MATH tasks for its high efficiency and superior performance. For the MCTS parameters, they are configured at different scales, as shown in Appendix A.6. We set $\beta_{\text{value}}$, $\beta_{\text{PRM}}$, and $\beta_{\text{ORM}}$ all to 1.0.

For policy self-improving (§4.5), we train the policy model up to 3 epochs, setting batch size to 128, learning rate to $5 \times 10^{-6}$ and minimal learning rate to $1 \times 10^{-6}$. Linear warm-up and decay is used with warm-up percent to be 10%. We perform early stopping based on a devset held out from the training instances. For GSM8K experiments, we perform two rounds of self-improving, synthesizing 6.4k and 7.9k prompts(Yu et al., 2023) respectively to obtain the corresponding MCTS outputs for training. For MATH experiments, we only perform one round of self-improving due to limited computation resources, and 5.9k prompts are synthesized.

The termination function for options can be either be learned or rule-based. In practice, for the GSM8K dataset, the termination condition occurs at the end of each line. This is based on the typical structure of this dataset, where each line represents a distinct step or point. For the MATH dataset, due to its complexity and the base model's tendency to generate many \n\n line breaks with some less meaningful content between them, termination occurs at the end of a line if a formula pattern is detected. During inference, if \n\n is encountered, we perform a rule-based check for formula patterns. It terminates if a pattern is found or continues generating until the next \n\n.

## 5.2   Results

Table 2 lists the performance comparisons of various methods on the GSM8K and MATH datasets. Our findings reveal that AʟᴘʜᴀLLM, based on Llama-2-70B and WizardMath-70B-V1.0, utilizes only final answer annotations and continues to improve through training on responses from $\eta$Mᴄᴛs. This comparison underscores the efficacy and broad applicability of our imagination-searching-criticizing self-improving framework. Moreover, when our model is augmented with $\eta$Mᴄᴛs decoding strategy, its performance markedly improves, achieving scores of 88.9 and 48.7 on the GSM8K and MATH datasets, respectively. Following two iterations of self-improvement using synthetic prompts, AʟᴘʜᴀLLM demonstrates performance comparable to that of GPT-4. This suggests a viable approach to improving LLMs' capabilities in complex problem-solving tasks in a self-improving fashion, leveraging a minimal amount of labeled data. We also analyze the performance of various search methods in Appendix A.8.

## 5.3   Ablation Study

We assess the effectiveness of each component in AʟᴘʜᴀLLM and report the results on GSM8K in Table 3(a). Vanilla MCTS, configured with only the value function and a fixed number of children per node, achieves an accuracy of 79.5%. This serves as a reference point for evaluating the incremental benefits introduced by each additional component. The use of adaptive branching increae the accuracy to 84.9%. The addition of `PRM` improves the accuracy modestly to 85.9%, showing the effectivenss of process supervision for searching. A more significant improvement is observed with the introduction of `ORM` with fast rollout, which boosts the accuracy to 86.5%. Integrating state merging results in

| Model | Decoding | #Annotation | RN | FA | SYN | GSM8K | MATH |
|---|---|---|---|---|---|---|---|
| GPT-3.5 | Sampling | - | - | - | - | 80.8 | 35.5 |
| GPT-4 | Sampling | - | - | - | - | 92.0 | 42.5 |
| GPT-4 (PAL) | Sampling | - | - | - | - | 94.2 | 51.8 |
| Gemini 1.0 Pro | Sampling | - | - | - | - | 77.9 | 32.6 |
| Gemini 1.0 Ultra | Sampling | - | - | - | - | 88.9 | 53.2 |
| Gemini 1.5 Pro | Sampling | - | - | - | - | 92.5 | 58.5 |
| Claude-2 | Sampling | - | - | - | - | 85.2 | 32.5 |
| PaLM-2 540B | Sampling | - | - | - | - | 80.7 | 34.3 |
| Llama-2-70b | Greedy | 0 | ✗ | ✗ | ✗ | 57.8 | - |
| Llama-2-70b SFT | Greedy | 7.5k | ✓ | ✓ | ✗ | 69.3 | - |
| WizardMath-70B-V1.0 | Greedy | 96k | ✓ | ✓ | ✗ | - | 20.7 |
| AlphaLLM | Greedy | 7.5k/7.5k | ✗ | ✓ | ✓ | 73.7 | 23.6 |
| AlphaLLM | $\eta$Mcts | 7.5k/7.5k | ✗ | ✓ | ✗ | 88.9 | 48.7 |
| AlphaLLM | $\eta$Mcts | 7.5k/7.5k | ✗ | ✓ | ✓ | 92.0 | 51.0 |

Table 2: Comparison results of AlphaLLM on the GSM8K and MATH datasets. #Annotation indicates the quantity of labeled data employed for fine-tuning policy or training critic models. The annotation used for training are noted as RN for rationales and FA for final answers. SYN means models trained on synthetic prompts, where trajectories were generated using $\eta$Mcts.

| AB | PRM | FR-ORM | SM | LG-#Rollout | Acc |
|---|---|---|---|---|---|
| ✗ | ✗ | ✗ | ✗ | ✗ | 79.5 |
| ✓ | ✗ | ✗ | ✗ | ✗ | 84.9 |
| ✓ | ✓ | ✗ | ✗ | ✗ | 85.9 |
| ✓ | ✓ | ✓ | ✗ | ✗ | 86.5 |
| ✓ | ✓ | ✓ | ✓ | ✗ | 87.0 |
| ✓ | ✓ | ✓ | ✓ | ✓ | 88.9 |

(a) Ablation study on GSM8K

| TA-ORM | Option | Acc | #Rollout |
|---|---|---|---|
| ✗ | ✗ | 38.8 | 201 |
| ✓ | ✗ | 44.1 | 198 |
| ✓ | ✓ | 45.4 | 148 |

(b) Ablation study on MATH

Table 3: **(a)**: Ablation studies on the GSM8K test set of various components of $\eta$Mcts, including adaptive branching, PRM, fast-rollout with ORM, state merge, and large number of rollouts. **(b)**: Ablation studies of the impacts of tool-augmented ORM and option-level formulation on MATH.

a further increase in accuracy, reaching 87.0%. Finally the combined of increasing the number of rollouts with the other components yields the best performance on this task.

Table 3(b) presents the ablation study of option formulation and the tool-augmented critic on the MATH dataset. Our proposed $\eta$Mcts achieves an accuracy of 45.4 with 148 rollouts. When options are excluded, reverting to essentially sentence-level MCTS, the performance decreases to 44.1 with a noticeable increase in the number of rollouts to 198. This demonstrates that option formulation introduces enhanced flexibility to MCTS, enabling better performance with fewer search efforts. Furthermore, the most significant decrease in performance is observed when only intrinsic knowledge is utilized for ORM, which drops to an accuracy of 38.8. This suggests that the absence of an external tool critically impedes the ORM's capability to effectively assess challenging math problems.

Figure 2 depicts a comparative results on GSM8K of two rounds of self-improving trained on trajectories collected using reranking and $\eta$Mcts. We report the performance of greedy decoding, $\eta$Mcts with a relatively small number of rollouts (50-60), and $\eta$Mcts with a larger number of rollouts (200-300) for each model. We observe that 1) Models trained on the trajectories from reranking or $\eta$Mcts outperform the initial policy by a significant margin. In addition, the performance can be iteratively improved with training suggesting that self-improving has the potential to achieve continual performance gain. 2) While both reranking and $\eta$Mcts can generate high-quality trajectories for self-improving , $\eta$Mcts is performant with high efficiency and better accuracy. Models trained on trajectories generated by it not only exceed the performance of those trained on reranked trajectories

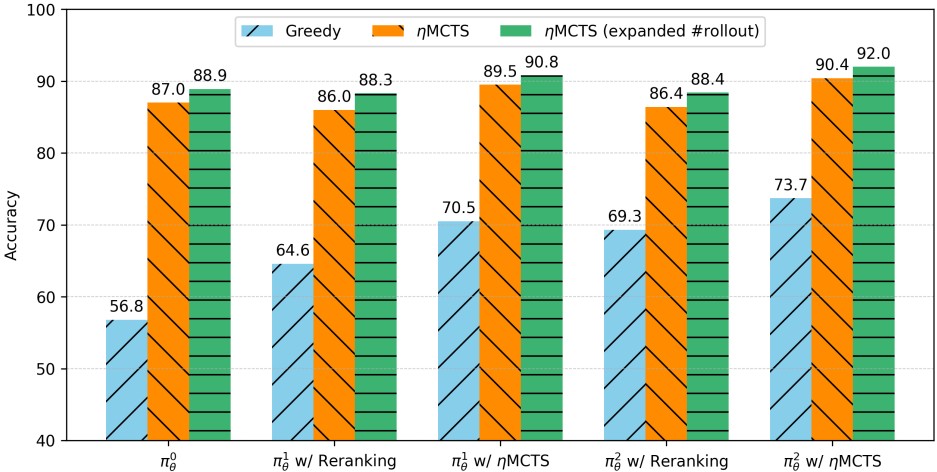

Figure 2: Empirical analysis on GSM8K of different self-improving data collection methods and number of iterations. Models are evaluated with greedy decoding, $\eta$MCTS with small #rollout and large #rollout.

but also, when decoded with $\eta$MCTS, demonstrate on par performance with GPT-4, revealing that ALPHALLM is an effective self-improving framework.

| Method | Threshold | Acc |
|---|---|---|
| Edit distance | 20 | 86.8 |
| Edit distance | 50 | 87.0 |
| Cosine Similarity | 0.7 | 86.3 |
| Model-based | N/A | 86.7 |

(a) Ablation on the choice of state merge functions.

| #Trajetory | Acc |
|---|---|
| 1 | 85.9 |
| 4 | 86.5 |
| 8 | 86.7 |

(b) Ablation on the number of trajectories.

Table 4: **(a)**: Ablation studies on the choice of heuristic/model-based functions in state merge on GSM8K with base Llama2-70b. The model used in the model-based state merge is Llama-2-70b-chat. **(b)**: Ablation studies of the number of rollout trajectories in fast-rollout estimation on GSM8K with base Llama2-70b.

We further analyze the impact of different hyperparameters and design choices for each component. Table 4(a) shows that varying heuristic functions (with hyperparameters) for state merge has limited impact on performance. Table 4(b) shows that, as the number of fast-rollouts increases, there is a corresponding improvement in performance. This is due to the reduction in the variance of the estimates. We used $n = 4$ in our experiments for better trade-off between performance and efficiency. Additional ablations on the choice of fast-rollout models, are provided in Appendix A.7.

## 6 Conclusion

In this paper, we introduce ALPHALLM, an imagination-searching-criticizing framework designed for the self-improvement of LLMs without the necessity of additional annotations. At the heart of it is the integration of MCTS with LLMs. To tackle the inherent challenges associated with this integration, including data scarcity, the vastness of search spaces, and the subjective nature of feedback in language tasks, we introduce a data synthesizer for strategic prompt synthesis, an optimized MCTS tailored for efficient search in language tasks, and a trio of critic models to provide precise feedback. Our experimental findings on mathematical reasoning tasks reveal that ALPHALLM significantly boosts the performance of LLMs without requiring extra data annotations. Moreover, when decoded with $\eta$MCTS, ALPHALLM performs comparably to GPT-4, highlighting the potential for self-improvement in LLMs.

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

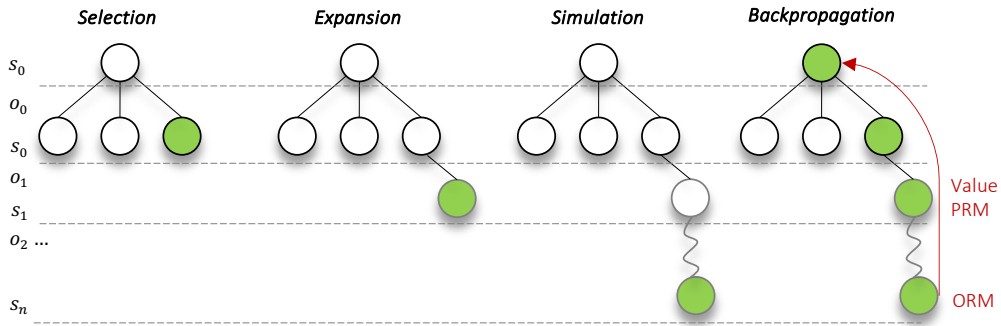

Figure 3: An overview of the four operations of $\eta$MCTS. A node is selected, expanded, simulated with fast rollout policy until a terminal node is reached, then the signals from value function, PRM and ORM are backpropagated.

# A Appendix

## A.1 Imagination, Searching, Criticizing and Learning Loop

---

**Algorithm 1:** LLM self-improving loop

---

**Input** Initial dataset $\mathcal{D}^0 = \{(\boldsymbol{x}_i^0, \boldsymbol{y}_i^0) \mid i \in [N]\}$, policy model $\pi_\theta^0$, reward model $R$, number of self-improving training loop $K$
**Output** $\theta^k$
**for** $k \leftarrow 1, \ldots, K$ **do**
  Generate synthetic prompts $[\boldsymbol{x}^k] = \text{SYN}(\pi_\theta^{k-1}, \mathcal{D}^{k-1})$
  Collect trajectories with search algorithm, *e.g.,* MCTS guided by $R$.
  $[\hat{\boldsymbol{y}}^k] = \text{MCTS}(\pi_\theta^{k-1}, [\boldsymbol{x}^k])$
  Construct dataset $\mathcal{D}^k = \{(\boldsymbol{x}^k, \hat{\boldsymbol{y}}^k)\}$
  Update policy $\theta^k = \arg\min_\theta L(\pi_\theta^{k-1}, \mathcal{D}^k)$
**end**

---

The algorithm is shown in Algorithm 1.

## A.2 Option-level MCTS

As illustrated in Figure 3, option-level MCTS consists of the following operations:

- **Selection** Starting from the root node, we iteratively select the child node based on Equation **??**.
- **Expansion** Once an expandable leaf node is selected, a new node is generated by starting with the previous state of the parent node as the initial option state. The option is then sampled using the policy $\pi$, and its completion is determined by the termination function $\beta$.
- **Simulation** The scaled reward of the newly expanded node, as well as some simulated future trajectories are evaluated using the feedback functions, which is discussed in §4.4.
- **Backpropagation** The average value of the newly generated node and all its ancestors is updated using the scaled reward from the evaluation step. Meanwhile, the visit counts for these nodes are also increased by one.

## A.3 Importance-Based Adaptive Branching Under Uniform Distribution

Let $V = \{v_\phi^\pi(\boldsymbol{s}_t, \boldsymbol{o}_t^1), v_\phi^\pi(\boldsymbol{s}_t, \boldsymbol{o}_t^2), ..., v_\phi^\pi(\boldsymbol{s}_t, \boldsymbol{o}_t^{m_t})\}$ be a set of $m_t$ values that are uniformly distributed. If the maximum and minimum values from $V$ are $v_{\max}$ and $v_{\min}$, the average gap between two consecutive values is given by $\frac{v_{\max} - v_{\min}}{m_t - 1}$. The upper bound of expected minimum distances from a new value $v_{\text{new}}$ to any value from $V$ is achieved when $v_{\text{new}}$ is consistently positioned at the midpoint between two consecutive values, and it is given by $\frac{v_{\max} - v_{\min}}{2(m_t - 1)}$.

Since $v_{\max} - v_{\min} = 2I(\boldsymbol{s}_t)$ for a uniform distribution, we can conclude that $E_\phi(t) \leq \frac{I(\boldsymbol{s}_t)}{m_t - 1}$.

**Theorem 4.1.** *The optimal branching factor $m_t$ in a tree search is set such that $m_t - 1$ is proportional to the node importance $I(\boldsymbol{s}_t)$, under the condition $\frac{I(\boldsymbol{s}_t)}{m_t - 1} \leq \epsilon$.*

*Proof.* We can have the optimization problem as:

$$\text{minimize:} \sum m_t$$
$$\text{subject to:} \frac{I(\boldsymbol{s}_t)}{m_t - 1} \leq \epsilon$$

Introduce the Lagrange multiplier $\lambda_t$ for each constraint:

$$L(m_t, \lambda_t) = \sum m_t + \sum \lambda_t \left(\epsilon(m_t - 1) - I(\boldsymbol{s}_t)\right)$$

Now, let's find the gradient of the Lagrangian with respect to $m_t$ and $\lambda_t$ and set them to zero:

$$\nabla_{m_t} L = 1 + \epsilon \lambda_t = 0$$
$$\nabla_{\lambda_t} L = \epsilon(m_t - 1) - I(\boldsymbol{s}_t) = 0$$

From the first equation, we get:

$$\lambda_t = -\frac{1}{\epsilon}$$

Substitute this value of $\lambda_t$ into the second equation:

$$\epsilon(m_t - 1) - I(\boldsymbol{s}_t) = 0$$

Solving for $m_t$, we get:

$$m_t = \frac{I(\boldsymbol{s}_t)}{\epsilon} + 1$$

Thus, $m_t - 1$ is proportional to the node importance $I(\boldsymbol{s}_t)$. $\qquad \square$

### A.4 Importance-Based Adaptive Branching Under Gaussian Distribution

If we assume that $v_\phi^\pi([\boldsymbol{s}_t, \boldsymbol{o}_t^j])$ and $v_\phi^\pi([\boldsymbol{s}_t, \boldsymbol{o}_t^i])$ are independent and identically distributed Gaussian random variables:

$$v_\phi^\pi([\boldsymbol{s}_t, \boldsymbol{o}_t^j]), v_\phi^\pi([\boldsymbol{s}_t, \boldsymbol{o}_t^i]) \sim \mathcal{N}(\mu, \sigma^2)$$

The difference $D_{ij} = v_\phi^\pi([\boldsymbol{s}_t, \boldsymbol{o}_t^j]) - v_\phi^\pi([\boldsymbol{s}_t, \boldsymbol{o}_t^i])$ will follow a normal distribution with:

$$D_{ij} \sim \mathcal{N}(0, 2\sigma^2)$$

To find the expected minimum absolute difference between $v_\phi^\pi([\boldsymbol{s}_t, \boldsymbol{o}_t^j])$ and the closest $v_\phi^\pi([\boldsymbol{s}_t, \boldsymbol{o}_t^i])$, we need to consider the distribution of the minimum of $m_t$ Gaussian differences.

The expected minimum value of $m_t$ absolute differences can be approximated using properties of order statistics for Gaussian distributions.

For a set of $m_t$ independent normal random variables with variance $2\sigma^2$, the expected minimum absolute difference, $\mathbb{E}[\min_i |D_{ij}|]$, can be approximated by:

$$E_\phi(t) \approx \frac{\sigma\sqrt{2}}{\sqrt{m_t}}$$

This approximation arises from the fact that the expected minimum value of the absolute deviations of normally distributed random variables scales with the inverse of the square root of the number of samples.

Then, assume the range of the $m_t$ samples are $R_m = max(v_\phi^\pi([s_t, o_t^i])) - min(v_\phi^\pi([s_t, o_t^i]))$, the the expected range $\mathbb{E}[R_m]$ of $m_t$ samples from a normal distribution can be approximated using properties of extreme values of Gaussian distributions. The range $R_m$ can be approximated as:

$$R_m \approx \sigma(z_{0.9995} - z_{0.0005})$$

where $z_p$ is the p-th percentile of the standard normal distribution. It can converge to

$$R_m \approx \sigma\sqrt{2\ln(m_t)}\left(2 - \frac{\ln(\ln(m_t))}{4\ln(m_t)}\right)$$

For simplicity, we can approximate the range using the primary term, which captures the dominant behavior:

$$R_m \approx \sigma\sqrt{2\ln(m_t)}$$

Then we have

$$E_\phi(t) \approx \frac{\sqrt{2}}{\sqrt{m_t}}\frac{R_m}{\sqrt{2\ln(m_t)}}$$

Knowing that for all distributions,

$$I(s_t) \geq \frac{R_m}{2}$$

We have

$$E_\phi(t) \leq \frac{I(s_t)}{\sqrt{m_t\ln(m_t)}}$$

Then to find the optimal $m_t$, the optimization problem is

$$\text{minimize:} \sum m_t$$

$$\text{subject to:} \frac{I(s_t)}{\sqrt{m_t\ln(m_t)}} \leq \epsilon$$

To solve this optimization problem, we can first rewrite the constraint in terms of $m_t$.

$$m_t\ln(m_t) \geq \frac{I^2(s_t)}{\epsilon^2}$$

Now, let's define a new function $g(m_t) = m_t\ln(m_t)$. We want to find the minimum $m_t$ such that $g(m_t) \geq \frac{I^2(s_t)}{\epsilon^2}$. To do this, we can find the derivative of $g(m_t)$ and set it to zero to find the critical points.

$$g'(m_t) = \frac{d}{dm_t}(m_t\ln(m_t)) = \ln(m_t) + 1$$

Setting the derivative to zero:

$$\ln(m_t) = -1$$

$$m_t = e^{-1}$$

However, this critical point corresponds to a minimum of the function $g(m_t)$, and we are interested in the minimum $m_t$ that satisfies the constraint $g(m_t) \geq \frac{I^2(s_t)}{\epsilon^2}$. Since the function $g(m_t)$ is increasing for $m_t > e^{-1}$, we can find the minimum $m_t$ by setting $g(m_t) = \frac{I^2(s_t)}{\epsilon^2}$ and solving for $m_t$:

$$m_t\ln(m_t) = \frac{I^2(s_t)}{\epsilon^2}$$

This can not be solved directly, but we can still observe that there is a positive correlation between $m_t$ and $I(s_t)$.

| Method | GSM8K | | MATH | |
|---|---|---|---|---|
| | Small | Large | Small | Large |
| $c$ | 1.0 | 1.5 | 1.0 | 1.0 |
| $\alpha$ | 1.0 | 1.0 | 1.0 | 1.0 |
| $c_{\max}(0)$ | 60 | 60 | 60 | 60 |
| $c_{\max}(t)$ where $t > 0$ | 10 | 10 | 10 | 10 |
| $c_{\min}(0)$ | 10 | 40 | 10 | 20 |
| $c_{\min}(t)$ where $t > 0$ | 2 | 2 | 3 | 3 |

Table 5: Parameters for MCTS. The Small/Large means small #rollout and small #rollout

## A.5 Prompt Templates

### A.5.1 PRM

> ###You are given a math problem, followed by a step-by-step reasoning process. Your task is to read the problem carefully, understand the solving steps, and check the correctness of the last reasoning step. Output 'True' if the last step is correct, and 'False' otherwise.\n\n### State\n{state}\n\n###Action\n{option}\n\n###Assessment\n{textual reward}

### A.5.2 ORM

> ###Assess a solution including final answer to a given math problem by following below steps.\n- Evaluate the method used for solving the problem.\n- Review each calculation step for accuracy. Check for computational errors, incorrect formula applications, or arithmetic mistakes.\n- The solution should use all the information provided in the question.\n- Examine the final answer for correctness, considering the calculations and method used.\n.\n\n###Prompt\n{prompt}\n\n###Trajectory\n{trajectory}\n\n###Assessment\n{textual reward}

### A.5.3 Policy Finetuning

For MATH experiments that take a WizardMath V1.0 70B as the policy, we adopt their proposed system prompt for self-improving. For GSM8K experiments taking Llama2 70B pretrain as the policy, we use the following system prompt.

> A chat between a curious user and an artificial intelligence assistant.\n The assistant gives helpful, detailed, and polite answers to the user's questions.\n User: $x_i$\n Assistant: $y_i$

## A.6 MCTS Details

We set the MCTS parameters in Table 5.

## A.7 Additional Ablations

**Fast-rollout model** Using Llama-2-70b instead of Abel-7B-002 improves performance by reducing bias from a smaller model, but Abel-002-7B is faster with similar computational resources due to higher concurrency and quicker processing. The details can be found in Table 6.

## A.8 Search Comparison

Table 7 presents the performance of various methods applied to different number of responses, from 10 to 50. Our analysis confirms several key findings: 1) Reranking utilizing ORM consistently outperforms self-consistency techniques, indicating that ORM is capable of generating meaningful

| Model | Acc (%) | Speed (s) |
|---|---|---|
| Abel-002-7B | 87.0 | 16.8 |
| Llama-2-70B | 87.3 | 38.1 |

Table 6: Ablation study over different fast-rollout models on GSM8K.

| Method | #Responses | GSM8K | | MATH | |
|---|---|---|---|---|---|
| | | #Rollouts | Accuracy | #Rollouts | Accuracy |
| Greedy | 1 | 4.6 | 57.8 | 9.9 | 20.7 |
| Self-consistency | 10 | 46 | 67.4 | 99 | 22.5 |
| | 30 | 137 | 74.2 | 299 | 27.3 |
| | 50 | 229 | 75.4 | 499 | 28.8 |
| Re-ranking | 10 | 46 | 80.8 | 99 | 34.1 |
| | 30 | 137 | 86.3 | 299 | 39.0 |
| | 50 | 229 | 87.7 | 499 | 42.0 |
| $\eta$MCTS | - | 55 | 87.0 | 223 | 45.4 |
| | - | 230 | 88.9 | 341 | 48.7 |

Table 7: Comparative results of various searching method on GSM8K and MATH.

signals for searching. 2) $\eta$MCTS demonstrates superior performance while requiring significantly fewer rollouts. For instance, on the MATH dataset, $\eta$MCTS achieves better results with only half the number of rollouts compared to reranking. Additionally, we evaluated the performance of BFS on the GSM8K only, where it requires 87.9 rollouts to achieve a score of 80.6. These results suggest that our design of an efficient MCTS in ALPHALLM can serve as an effective policy improvement operation, enabling the search for high-quality trajectories with reduced computational cost.

## A.9 Rollout Example

Consider the following GSM-like question:

> Question: Sandy's monthly phone bill expense is equal to ten times her age now. In two years, Sandy will be three times as old as Kim. If Kim is currently x years old, calculate Sandy's monthly phone bill expense.\nIf we know the answer to the above question is 340, what is the value of the unknown variable x?\n

A node in the second layer could have the following content:

> Answer: We know that Sandy's monthly phone bill is 10 times her age. In two years, Sandy will be 3 times as old as Kim. The sum of Sandy's age now and 2 years is 3 times the sum of Kim's age now and two years.\nSandy's age now is 340/10 = «340/10=34»34. In two years, Sandy's age will be 34 + 2 = «34+2=36»36.\n

The parent of this node has the content:

> Answer: We know that Sandy's monthly phone bill is 10 times her age. In two years, Sandy will be 3 times as old as Kim. The sum of Sandy's age now and 2 years is 3 times the sum of Kim's age now and two years.\n

And one of its fast-rollout paths could be:

The sum of Sandy's age now and 2 years is 36. The sum of Kim's age now and two years is x + 2.\n36 = 3(x + 2)\n6 = 3x + 6\n3x = 30\nx = 10\n #### 10

## A.10 Critic Performance

We evaluated the performance of the value function and `PRM` on the GSM8K test set. Table 8 presents a comparison of these models in terms of precision, recall, and Expected Calibration Error (ECE). Results indicate that the value function achieves higher precision and better calibration, while `PRM` demonstrates a superior recall.

| Model | Precision | Recall | ECE |
|---|---|---|---|
| Value Function | 0.82 | 0.79 | 0.032 |
| PRM | 0.62 | 0.90 | 0.375 |

Table 8: Performance comparison of the Value Function model and `PRM` on the GSM8K test set.

## A.11 Compute Resources

Our experiments were conducted using NVIDIA A100 40GB GPUs. Serving models based on Llama-2-70B or WizardMath-70B required 4 GPUs, while serving Llama-2-7B and Abel-002-7B was possible on a single GPU. Training the 70B models required 64 GPUs.

## A.12 Limitations and Future Work

Despite the promising results demonstrated by ALPHALLM in this study, there are several limitations that requires further exploration. (*i*) Our current implementation employs relatively simple methods for generating synthetic prompts. Future iterations of ALPHALLM should explore advanced techniques, such as Self-Instruct, to create both diverse and model capability-awared prompts. (*ii*) Although ALPHALLM demonstrates improvements over base models, its performance in greedy sampling is substantially inferior to that observed when decoded with $\eta$MCTS. This indicates that the full potential of MCTS for self-improvement in LLMs has not yet been fully realized. Two potential factors contributing to this issue have been identified: a) the self-improvement loop may not be leveraging sufficient data; and b) the base model may be limited in its capacity for rapid learning. Addressing these concerns could lead to more significant improvemens. (*iii*) In our existing framework, the critic models remain static. We will explore mechanisms to continually update critic models to adapt to new policy models. This will help ensure the discriminator-generator gap and improve the overall training dynamics. (*iv*) The evaluation of ALPHALLM has been limited to mathematical reasoning tasks. To verify the generalizability and broader applicability of the framework, future research will need to extend its application to other domains.

