# OpenReview forum: "Toward Self-Improvement of LLMs via Imagination, Searching, and Criticizing"
_NeurIPS.cc/2024/Conference — NeurIPS 2024 poster_

### Official Review · Reviewer_iEWd · 2024-07-08

**Soundness:** 3
**Presentation:** 2
**Contribution:** 3
**Rating:** 5
**Confidence:** 3

**Summary:**

This paper introduces ALPHALLM, which is an imagination-searching-criticizing framework designed for self-improvement. Inspired by AlphaGo, authors integrate MCTS and LLMs to establish the self-improvement loop. Additionally, authors proposed eta-MCTS, which is a decoding method used to reduce the search space. The experiments show that ALPHALLM can increase LLMs’ reasoning ability obviously, especially combined with eta-MCTS decoding.

**Strengths:**

1. It’s innovative to use tree search to do self-improvement and it’s good to identify some challenges very clearly.
2. The experiments show that the reasoning ability of LLMs can be improved effectively.

**Weaknesses:**

1. It could be more clear in some parts and including some examples would be helpful. For example, in the part of 4.2 data synthesizing, I’m curious about how the synthesized data looks like.
2. Authors identify that one of the challenges in working on LLMs’ self-improvement is the difficulty to get clear evaluations, and they proposed some method to get the evaluations, i.e. value function, PRM, ORM. But they do not discuss whether these methods are reliable to get perfect feedback.

**Questions:**

Please refer to the part of weaknesses.

**Limitations:**

Limitations are sufficiently discussed.

---

> ### Author Rebuttal · Authors · 2024-08-05
>
> We greatly appreciate the reviewer’s insightful feedback. Your recognition of our novelty, clear identification of challenges, and the effectiveness of our results is is highly encouraging to us.
>
> ---
>
> > **[W1]** It could be more clear in some parts and including some examples would be helpful. For example, in the part of 4.2 data synthesizing, I’m curious about how the synthesized data looks like.
>
> We understand that some sections of the paper, such as the data synthesizing part in section 4.2, could benefit from further clarification and examples.
> To address your query about the synthesized data, we used the method described in [1] to generate synthesized questions. Here's an example synthesized based on GSM8K:
> ```
> Question: Sandy's monthly phone bill expense is equal to ten times her age now. In two years, Sandy will be three times as old as Kim. If Kim is currently x years old, calculate Sandy's monthly phone bill expense.\nIf we know the answer to the above question is 340, what is the value of unknown variable x?\n
> ```
> We hope this example provides a clearer understanding of how our data synthesizing process works. We are open to further discussions to improve the clarity of our work.
>
> [1]. Yu, Longhui, et al. "Metamath: Bootstrap your own mathematical questions for large language models." arXiv preprint arXiv:2309.12284 (2023).
>
> ---
>
> > **[W2]** Authors identify that one of the challenges in working on LLMs’ self-improvement is the difficulty to get clear evaluations, and they proposed some method to get the evaluations, i.e. value function, PRM, ORM. But they do not discuss whether these methods are reliable to get perfect feedback.
>
> We appreciate your insightful comment regarding the reliability of the evaluation methods such as the value function, PRM, and ORM for obtaining ideal feedback in the context of LLMs' self-improvement. In our study, we have utilized a combination of these evaluation methods to ensure better calibration results.
> The accuracy obtained on the GSM8K dataset for each critics is as follows:
> - Value: 0.832
> - PRM: 0.841
> - ORM: 0.877
>
> As a combination of them, we have achieved the following calibration scores:
> - ECE: 0.1319
> - AUROC: 0.958
> - Brier Score: 0.0429
>
> These results indicate that our proposed evaluation methods have provided reasonably good feedback for LLMs' self-improvement. We appreciate the suggestion and will continue to explore more reliable and robust evaluation techniques to enhance the self-improvement capabilities of LLMs in future research.

---

> > ### Comment · Reviewer_iEWd · 2024-08-13
> >
> > Thanks for the reply and these address some of the questions. I will maintain my score.

---

> > > ### Author Response · Authors · 2024-08-14
> > > **Thank you for your feedback**
> > >
> > > Thank you and we appreciate your feedback.

---

### Official Review · Reviewer_LHqU · 2024-07-13

**Soundness:** 3
**Presentation:** 3
**Contribution:** 3
**Rating:** 6
**Confidence:** 3

**Summary:**

This paper proposes an imagination-searching-criticizing approach called ALPHALLM to enhance the capabilities of large language models (LLMs). ALPHALLM integrates Monte Carlo Tree Search (MCTS) with LLMs to establish a self-improving framework.

**Strengths:**

- This paper introduces a novel approach to LLM self-improvement using MCTS, presenting an interesting concept.
- The paper is well-written, and the proposed imaginative-searching-criticizing approach is clearly explained.
- Using only 7.5k/3k final math answer annotations and after just two iterations of self-improvement, ALPHALLM achieves impressive results: 92.0 on GSM8K and 51.0 on MATH. These results are remarkable.

**Weaknesses:**

- Given only final math answer annotations, ALPHALLM essentially performs the final-label (reward) classification problem and tree search based on the predicted reward to generate rationales for each final answer. It is still unclear how ALPHALLM could outperform other LLM-SFTs that utilize both rationale annotations and final answer annotations.
- ALPHALLM is a general framework. The authors should apply ALPHALLM to other tasks where learning signals are clear to demonstrate the overall effectiveness of this self-improvement framework.

**Questions:**

- Have you ever tried using your value/reward function to perform RL fine-tuning with PPO?

**Limitations:**

None.

---

> ### Author Rebuttal · Authors · 2024-08-05
>
> Thank you for your thoughtful feedback and insightful questions. We are encouraged by your approval of the novelty and effectiveness of ALPHALLM, as well as the clarity of this approach.
>
> ---
>
> > **[W1]** Given only final math answer annotations, ALPHALLM essentially performs the final-label (reward) classification problem and tree search based on the predicted reward to generate rationales for each final answer. It is still unclear how ALPHALLM could outperform other LLM-SFTs that utilize both rationale annotations and final answer annotations.
>
> Our primary motivation stems from the observation that as LLMs continue to evolve, achieving human parity in numerous tasks, the necessity and quality of explicit rationale annotations in datasets such as GSM8K and MATH become points of consideration.
> - The rationale annotations provided in datasets like GSM8K and MATH, while useful, are not necessarily optimal or superior to those that can be generated through advanced methods like MCTS guided approaches. The quality and suitability of these human-provided rationales can vary, and they might not always align with the nuanced reasoning capabilities of advanced LLMs. Our hypothesis is that the rationales generated by ALPHALLM, guided by MCTS, are potentially more robust and aligned with the model’s training process than those annotated in GSM8K and MATH.
> - ALPHALLM utilizes final math answer annotations in conjunction with tree search based on predicted rewards to derive rationales. This method allows the model to explore a wider range of reasoning paths and select the most plausible ones based on calculated rewards, rather than being confined to potentially suboptimal human-provided annotations.
> - As presented in our experimental results, ALPHALLM demonstrates superior performance compared to WizardMath, which utilizes over 96k data points for math-specific training, including both rationale and final answer annotations
>
> ---
>
> > **[W2]** ALPHALLM is a general framework. The authors should apply ALPHALLM to other tasks where learning signals are clear to demonstrate the overall effectiveness of this self-improvement framework.
>
> We appreciate the reviewer’s suggestion regarding the application of ALPHALLM to a broader range of tasks. We are actively working on extending our framework to other STEM reasoning tasks to further validate its effectiveness. Preliminary results are promising, and we are committed to sharing a comprehensive report on these additional tasks in the near future!
>
> ---
>
> > **[Q1]** Have you ever tried using your value/reward function to perform RL fine-tuning with PPO?
>
> We have not yet explored vanilla PPO but optimizing LLMs using trajectories from ALPHALLM with PPO/DPO is indeed on our roadmap for future work. To provide a comparative analysis, we included the results of Best-of-N ranked by our ORM in the current study. Best-of-N is often considered an alternative to PPO, particularly when N is sufficiently large. As demonstrated in Table 5 in the submission, ALPHALLM significantly outperforms Best-of-N, highlighting the effectiveness of our MCTS approach. This substantial margin of improvement underscores the potential of our method over traditional approaches like PPO, even before we integrate them.

---

> ### Comment · Reviewer_LHqU · 2024-08-13
>
> Thank you for your response. Below is my feedback:
>
> - To support the hypothesis that the rationales generated by ALPHALLM are more robust and aligned, the author should provide additional examples of these rationales and analyze the differences between human-annotated rationales and those generated by ALPHALLM. Without this, the theoretical strength of ALPHALLM remains unclear.
> - After reading comments from Reviewers SX7Z and PPNK, I agree with their concerns about the implementation details. I also believe that the code should be provided to allow for verification of each component of your method.
>
> Due to the main concerns raised above, I have decided to lower my score. However, given that the experimental results are indeed remarkable, I still support the acceptance of this paper.

---

> > ### Author Response · Authors · 2024-08-14
> > **Thank you for your feedback**
> >
> > Thank you for the constructive feedback! We agree that a specific analysis between human-annotated rationales with those generated by ALPHALLM is valuable. Indeed, we have already provided evidence in the fine-tuning results. For instance, in Table 2 (around line 313), we show that ALPHALLM, trained with self-generated rationales, outperforms LLaMA-2 70B SFT, which uses human-annotated rationales. This empirical evidence could support the hypothesis that the rationales generated by ALPHALLM are more effective. We also agree that providing additional examples of these rationales and analyze the differences could provide deep insights. We will definitely include this analysis in the appendix in the next version.
> >
> > Regarding the implementation details, we have included additional ablations and details in our response to Reviewer PPNK and Reviewer SX7Z. We kindly ask you to review those additional results to see if they address your concerns. We understand that some details (e.g. Appendix A.6) might be overlooked without a thorough review of the appendix. To further ensure the reproducibility of our results and to facilitate the verification of each component of our method, we are also committed to providing the codebase associated with ALPHALLM. Thank you for your understanding!

---

### Official Review · Reviewer_PPNk · 2024-07-30

**Soundness:** 3
**Presentation:** 2
**Contribution:** 3
**Rating:** 6
**Confidence:** 5

**Summary:**

The paper proposes AlphaLLM, a tree-search enhanced framework with a few improvements over Data Synthesizing, option-level MCTS, Importance-Based Adaptive Branching, state merging, fast rollout, critic function and policy improvement process. Experimental results verify the framework's effectiveness on GSM8k and MATH.

**Strengths:**

1. The technical contribution seems solid. The paper proposes a comprehensive framework and address modifications/improvements over the full pipeline of tree-search.
2. Experimental results demonstrate great potentials for the proposed algorithms.

**Weaknesses:**

1. The writing is not clear enough. It is not that clear to me how the option-level is implemented to separate sentences (how the Termination function is determined).
2. There are many different components in the pipeline design but the ablation studies are not enough to validate them one by one. For example, It at least need to involve:
2.1 Comparison with other heuristic search methods, for example: (1) the beam-search + value function described in [1, 2]) (2) majority-vote or reranking with ORM.
2.2 Wall time or token consumption comparison between different methods.
2.3 The ablation studies seem to be too simple to present how different components can really influence the performance. Pure with/without results seem to be too simple. More experiments like how to control the hyperparameter for these things are required. For example, how different heuristic functions can influence the State Merge, how different fast rollout model can influence the effiency, how the thresold in ORM in the process of data generation can influence the improvement process, etc.


Reference
[1] Feng, Xidong, et al. "Alphazero-like tree-search can guide large language model decoding and training." arXiv preprint arXiv:2309.17179 (2023).
[2] Yu, Fei, Anningzhe Gao, and Benyou Wang. "Outcome-supervised verifiers for planning in mathematical reasoning." arXiv preprint arXiv:2311.09724 (2023).

**Questions:**

1. What is the difference between value function training and PRM training? From the description it seems that the PRM's training is exactly the same as value function training in Monte-Carlo estimation (except you tend to take more simulations starting from a fixed state).

Also, it seems that the technique described in the critic section has almost been described in many previous works: value function training in [1], PRM training in [2, 3], and ORM in [4], the author needs to clarify more on their own contributions and make proper citations.

2. One difference between your work and [1] is that they leverage the value function itself to backward (like AlphaZero) while you are using the fast model to rollout to reach the terminal state, which is like the initial version of AlphaGo (where they have a fast-move network) and [5]. What do you think of these differences and why do you make this design choices?


Reference
[1] Feng, Xidong, et al. "Alphazero-like tree-search can guide large language model decoding and training." arXiv preprint arXiv:2309.17179 (2023).
[2] Wang, Peiyi, et al. "Math-shepherd: A label-free step-by-step verifier for llms in mathematical reasoning." arXiv preprint arXiv:2312.08935 (2023).
[3] Lightman, Hunter, et al. "Let's verify step by step." arXiv preprint arXiv:2305.20050 (2023).
[4] Uesato, Jonathan, et al. "Solving math word problems with process-and outcome-based feedback." arXiv preprint arXiv:2211.14275 (2022).
[5] Hao, Shibo, et al. "Reasoning with language model is planning with world model." arXiv preprint arXiv:2305.14992 (2023).

**Limitations:**

See questions and weaknesses.

---

> ### Author Rebuttal · Authors · 2024-08-07
>
> >  **[W1]** It is not that clear how the option-level is implemented to separate sentences
>
> Thank you for your feedback regarding the clarity of our writing. To clarify, the termination function for options operates differently depending on the dataset:
> - For the GSM8K dataset, the termination condition occurs at the end of each line. This is based on the typical structure of this dataset where each line represents a distinct step or point.
> - For the MATH dataset, due to its complexity and the base model's tendency to generate many '\n\n' line breaks with some less meaningful content between them, termination occurs at the end of a line if a formula pattern is detected. During inference, if '\n\n' is encountered, we perform a rule-based check for formula patterns. It terminates if a pattern is found or continues generating until the next '\n\n'.
>
>
> We will include the clarifications in the revision.
>
> ---
>
> >  **[W2.1, W2.2]**  The ablation studies are not enough. 2.1 Other heuristic search methods; 2.2 Token consumption
>
> In Table 5 of the original submission, a basic comparison of accuracy and the number of rollouts for self-consistency (majority-vote) and re-ranking methods for both GSM8K and MATH datasets was included. Following the suggestion, beam search (BFS) has also been incorporated into the experiments on GSM8K, as shown in the table below:
> Method | #Responses | #Tokens | Acc
> ---|---|---|---
> Greedy | 1 | 127 | 57.8
> Maj-Vote | 10 | 1272 | 67.4
> Maj-Vote | 30 | 3788 | 74.2
> Maj-Vote | 50 | 6332 | 75.4
> Re-ranking | 10 | 1272 | 80.8
> Re-ranking | 30 | 3788 | 86.3
> Re-ranking | 50 | 6332 | 87.7
> BFS | - | 2430 | 80.6
> \etaMCTS | - | 1521 | 87.0
> \etaMCTS | - | 6360 | 88.9
>
> Token consumption is estimated by tracking MCTS rollouts and multiplying by the average tokens per rollout for Llama2-70b on GSM8K. The table indicates that \etaMCTS outperforms and is more efficient than other baselines.
>
> ---
>
> > **[W2.3]** 2.3 The ablation studies seem to be too simple
>
> In addition, we have conducted ablation studies on GSM8K with base Llama2-70b to further demonstrate the influence of various components on the performance:
>
> **State Merge**: The impact of the choice of heuristic functions (w/ hyperparameters) or model-based state merge is not very significant.
>
> Method | Threshold | Acc
> ---|---|---
> Edit distance | 20 | 86.8
> Edit distance  | 50 | 87.0
> Cosine Similarity (TF-IDF) | 0.7 | 86.3
> Model-based (Llama-2-70b-chat )| N/A	 | 86.7
>
> **Fast-rollout**:  Number of rollouts: As the number of fast-rollouts increases, there is a corresponding improvement in performance. This is due to the reduction in the variance of the estimates. We used n=4 in our experiments for better trade-off between performance and efficiency.
> #rollout| Acc
> ---|---
> 1|85.9
> 4|86.5
> 8|86.7
>
> **Fast-rollout model**: Using Llama-2-70b instead of Abel-7B-002 improves performance by reducing bias from a smaller model, but Abel-002-7B is faster with similar computational resources due to higher concurrency and quicker processing.
> Model | Acc| Speed (s)
> ---|---|---
> Abel-002-7B|87.0|16.8
> Llama-2-70b|87.3|38.1
>
> **ORM**: During training, each question is sampled multiple times, and the ORM predicts the correctness of each trajectory as True or False. The ORM's score (probability of outputting True) is used in the data generation process, along with the scores from value and PRM, eliminating the need for a threshold.
>
> **Adaptive branching**: with adaptive branching, the performance score is 84.9, whereas without it, and using a fixed number of 4 branches, the score drops to 79.5.
>
> ---
>
> > **[Q1]** What is the difference between value function and PRM training? Also, please clarify your contributions and provide proper citations in the critic section.
>
> Value function training and PRM training share some similarities, but they differ in task formulation, model architecture, and training loss.
>
> Value function training focuses on estimating the expected FUTURE return of a given state or state-action pair. In contrast, PRM evaluates the quality of the current state or node. While PRM ideally requires quality labels for each state, due to the high cost and time involved in obtaining these, MC estimation is used as a proxy.
>
> The Value Function model extends from a policy model by adding a value head and is trained using regression loss to predict expected returns, a continuous variable. PRM, trained using LM loss, evaluates the quality of states/nodes and benefits from the contextual understanding of language models. The differences in architecture and training result in distinct behaviors: the Value Function model has better precision (0.82 vs. 0.62) and calibration (ECE: 0.032 vs. 0.375), while PRM has superior recall (0.90 vs. 0.79).
>
> Finally, our contribution lies in utilizing a trio of critics, each with strengths in different aspects, to provide reliable signals for guiding MCTS. We will restructure the critic section to clarify these points further, and will also ensure to appropriately cite the references [1,2,3,4] in that section to acknowledge their contributions to this field.
>
> ---
>
> > **[Q2]** Why choose to include the fast-rollout?
>
> We use rollouts to the terminal state, because empirical evidence shows that the value network's accuracy improves with detailed information closer to the terminal state. For example, an early version of the value network had 89.5% accuracy at the terminal state and 83.2% accuracy three steps before, indicating more reliable predictions near the terminal state. Estimations at the end of rollout trajectories have lower bias but higher variance, so using the mean score from multiple rollouts offers low-bias, low-variance estimations. To mitigate bias from fast rollouts of a smaller model, we also keep value estimations at the actual progress point. Additionally, using the outcome reward model through fast rollouts provides extra signals that enhance learning, making our approach more robust and effective.

---

> ### Author Response · Authors · 2024-08-07
> **Rebuttal Supplement: References**
>
> Due to the word limit, we included the references used in the rebuttal for reviewer PPNk in this comment.
>
> We would also like to greatly thank the reviewer PPNk for the valuable feedback and insightful questions. We appreciate the recognition of our technical contributions and the potential of our algorithm. The reviewer's questions and suggestions have been crucial in improving the clarity of our work.
>
> ## Reference
>
> [1] Feng, Xidong, et al. "Alphazero-like tree-search can guide large language model decoding and training." arXiv preprint arXiv:2309.17179 (2023).
> [2] Wang, Peiyi, et al. "Math-shepherd: A label-free step-by-step verifier for llms in mathematical reasoning." arXiv preprint arXiv:2312.08935 (2023).
> [3] Lightman, Hunter, et al. "Let's verify step by step." arXiv preprint arXiv:2305.20050 (2023).
> [4] Uesato, Jonathan, et al. "Solving math word problems with process-and outcome-based feedback." arXiv preprint arXiv:2211.14275 (2022).

---

> > ### Comment · Reviewer_PPNk · 2024-08-13
> >
> > Despite the difference on architecture and training loss, the clarification on formulation difference makes me still confused:
> >
> > "Value function training focuses on estimating the expected FUTURE return of a given state or state-action pair. In contrast, PRM evaluates the quality of the current state or node. While PRM ideally requires quality labels for each state, due to the high cost and time involved in obtaining these, MC estimation is used as a proxy."
> >
> > Since you are using MC estimation, then this PRM is exactly the same for estimating the expected future return of a given state or state-action pair, so it should fall back to value function training.

---

> > > ### Author Response · Authors · 2024-08-13
> > > **Thank you for your feedback**
> > >
> > > Thank you for your insightful comment! We agree that the training objective of the MC proxy is indeed the same as that of the value function, and both the estimated values should tend to be the same as long as the MC samples and data used for training the value function go to infinity. The reason we adopted this approach is due to the high cost and time involved in obtaining quality labels like [1] for each state. Therefore, we adapted the PRM used in [2,3] as a proxy .
> > >
> > > In practice, this approach has proven to be useful. The value function exhibits better precision and calibration, while PRM has superior recall. By integrating these models, we observed an overall performance improvement from 84.9% to 85.9% on GSM8K.
> > >
> > > In future work, we plan to explore additional methods to obtain more accurate estimations of the quality of the current step. We appreciate your valuable feedback and thank you once again for your contribution to our research.
> > >
> > >
> > > [1] Lightman, Hunter, et al. "Let's verify step by step." arXiv preprint arXiv:2305.20050 (2023).
> > > [2] Wang, Peiyi, et al. "Math-shepherd: A label-free step-by-step verifier for llms in mathematical reasoning." arXiv preprint arXiv:2312.08935 (2023).
> > > [3] Jiao, Fangkai, et al. "Learning planning-based reasoning by trajectories collection and process reward synthesizing." arXiv preprint arXiv:2402.00658 (2024).

---

> > > > ### Comment · Reviewer_PPNk · 2024-08-13
> > > >
> > > > I am satisfied by the response, but do remember to include these explanation to make things more clear in the main paper. I will raise my score to 6.

---

> > > > > ### Author Response · Authors · 2024-08-14
> > > > > **Thank you for your feedback**
> > > > >
> > > > > Thank you for your positive feedback. We will ensure that these explanations are included in the main paper to enhance clarity.

---

### Official Review · Reviewer_SX7Z · 2024-07-31

**Soundness:** 2
**Presentation:** 3
**Contribution:** 3
**Rating:** 6
**Confidence:** 3

**Summary:**

The paper introduces a method for self-imporvement of LLMs called AlphaLLM. The method consists of three components:
Generation of expert trajectories
Effective Monte-Carlo-Tree-Search over the LLM outputs (nablda-MCTS)
A series of critics for evaluating reliable reward signal (value function, Process Reward Model, Outcome Reward Models)

The method can be used as a rollout mechanism to generate high quality outputs. To help deal with the large branching factor, the authors propose three subcomponents:
State space is set to Options, which is neither token space nor sentence space, but a variable length sequence which terminates based on an auxiliary heuristic beta.
States are merged using another auxiliary function p_vM
Fast rollouts are used with a smaller LLM to help estimate.

The authors then demonstrate incredibly strong empirical results on both MATHS and GSM8K. Training with the proposed method on LLAMA-2-70b is able to achieve 51% on the MATH dataset.

**Strengths:**

The method seems to work well!

The discussion on state merge and options helps express difficulties in applying MCTS.

The paper seems to combine many subcomponents to produce a strong algorithm.

The branching factor is well-motivated.

**Weaknesses:**

Lots of experimental details are missing. There is very little discussion of how AlphaLLM is trained or what base model it is (it's only mentioned in the introduction!). In particular, no information is given about the actual training of either the policy (LLM), the Critics, or the heuristic functions p_vM.

The explanation of the options is lacking - even looking at A.2 the explanation of options is unclear. Is Beta learnt as a function or generated in some other way? Are the set of options I, learnt before a rollout or generated on the fly?

Ablations are missing - in particular using different state methods, removing the Importance-Based Adaptive Branching, State Merge or Option-Level MCTS.

The prompts in the appendix are reduced so we can’t replicate.

No Code provided.

Weird choice of baselines - surely to keep evaluations consistent with other LLMs, you should allow models to generate the same amount of output tokens with access to the PRM or ORM? I find it hard to believe COT on Claude-2 is a fair comparison, when the amount of optimisation pressure applied to multiple components of the method (including gradient updates).


There is no example of a rollout provided.

**Questions:**

- What policy was used in the end for the fast rollout?
- The prompt templates are missing key parts - [A detailed rubric that specifies how to evaluate a step of a task] would be helpful if shared. - In particular when comparing to GPT-4 or other model perform was an equally informative prompt used? (Or at least a prompt of equal input tokens).
- If the synthetic data generated is used few-shot by llama2, how does your model perform?
- I think the related work is missing some key papers such as: [https://arxiv.org/abs/2203.14465] (the STAR family of papers)

**Limitations:**

The paper in its current form is not replicable.

---

> ### Author Rebuttal · Authors · 2024-08-05
>
> Your suggestions have helped us a lot to improve the work, though we do think there might exist misunderstandings about our method. We hope the following clarifications and additional evidences would be possible for a re-evaluation.
>
> ---
>
> > **[W1]** Lots of experimental details are missing. There is very little discussion of how AlphaLLM is trained or what base model it is (it's only mentioned in the introduction!). In particular, no information is given about the actual training of either the policy, the Critics, or the heuristic functions p_vM
>
> Model and training details: As detailed in **Appendix A.6** (line 609-624) in the original submission, the base models used are Llama-2-70b for GSM8K and WizardMath-70B-V1.0 for MATH, respectively. More details regarding the training (LR, warm-up, etc.), critics (data and model), and the fast-rollout are also provided in **A.6**.
>
> Heuristic functions: As mentioned in **Sec 4.3.3**, the heuristic function p_vM can either be a faster rule-based measurement (e.g. edit distance) or a model-based method (e.g. prompting a LLM). For our experiments, we used edit distance. We also include additional ablation studies demonstrating that the performance is not very sensitive to the choice of p_vM:
> Method | Threshold | Acc
> ---|---|---
> Edit distance | 20 | 86.8
> Edit distance  | 50 | 87.0
> Cosine Similarity (TF-IDF) | 0.7 | 86.3
> Model-based (Llama-2-70b-chat )| N/A     | 86.7
>
> ---
>
> > **[W2]** The explanation of the options is lacking. Is Beta learnt or generated in some other way? Are the set of options I, learnt before a rollout or generated on the fly?
>
> The termination function $\beta$ can be either be learnt or rule-based. In practice, for the GSM8K, we used a newline character (\n) as the termination condition, as shown in **Table 4**. For the MATH, termination condition occurs when a line with a formula pattern is found, with line breaks indicated by '\n\n'. During inference, if the generated output has '\n\n', we check for formula patterns using rules. It terminates if a pattern is detected or continues until another '\n\n' appears.
>
> The set $I \subseteq S$  represents a set of initial states (not a set of options). It can be the state of a question or a previously terminated step.
>
> ---
>
> > **[W3]** Ablations are missing - in particular using different state methods, removing the Adaptive Branching, State Merge or Option-Level MCTS.
>
> We have included ablations for different state methods, including PRM, fast-rollout with ORM, state merge, and large number of rollouts, tool-augmented ORM and option-level MCTS in **Table 3**, with a discussion in **Sec 5.3**. Additional ablations on adaptive branching are included here:
>   - With adaptive branching: 84.9
>   - Without adaptive branching: 79.5
>
> ---
>
> > **[W4]** No Code provided.
>
> We plan to release the code after it undergoes a legal review by our entity.
>
> ---
>
> > **[W5]** The prompts in the appendix are reduced so we can’t replicate.
>
> All the prompt templates will be included with the code release. These templates are detailed in Appendix A.5 , with specific rubrics for PRM and ORM are provided in Section A in the comment below.
>
> ---
>
> > **[W6]** Weird choice of baselines - surely to keep evaluations consistent with other LLMs, you should allow models to generate the same amount of output tokens with access to the PRM or ORM? I find it hard to believe COT on Claude-2 is a fair comparison, when the amount of optimisation pressure applied to multiple components of the method.
>
> Our primary focus in this paper is to explore the self-improvement capabilities of LLMs. The experimental results demonstrate that our method, AlphaLLM, can achieve significant self-improvement. Moreover, we observed that the performance of Llama-2-70b, when employs our proposed \etaMCTS, is comparable to that of Claude-2 and GPT-4, highlighting the potential for self-improvement.
>
> We included proprietary models mainly to illustrate the potential of our self-improvement method. The definition of a "fair comparison" can vary. Proprietary models, such as the latest versions of Claude-2 and GPT-4, are trained with unknown data quantities and types, often iteratively refined. By default, these models already incorporate chain-of-thought processes. It is plausible that proprietary models leverage System-2 results to enhance System-1 outputs, complicating direct comparisons.
>
> In addition, we have demonstrated superior performance when compared with recent work [1] that also used MCTS and utilizes a similar token usage amount. For example, AlphaLLM achieved 51.0 on MATH while [1] achieved 34.3.
>
> We hope this clarifies our approach and rationale behind the baseline choices. We welcome further suggestions to improve our evaluation methodology.
>
> ---
>
> > **[W7]** There is no example of a rollout provided.
>
> Please refer to the example in the Section B in the comment.
>
> ---
>
> > **[Q1]** What policy was used in the end for the fast rollout?
>
> As mentioned in **Appendix A.6**, the fast-rollout model is Abel-002-7B.
>
> ---
>
> > **[Q2]** The prompt templates are missing key parts - In particular when comparing to GPT-4 or other model perform was an equally informative prompt used? (Or at least a prompt of equal input tokens).
>
> Please refer to the responses for [W5] and [W6].
>
> ---
>
> > **[Q3]** If the synthetic data generated is used few-shot by llama2, how does your model perform?
>
> The synthetic data would still have the same question format as [2] but with answer steps from MCTS. Therefore, Llama2 with few-shot prompting would exhibit similar performance to that observed on the questions in [2].
>
> ---
>
> > **[Q4]** I think the related work is missing some key papers such as the STAR family of papers
>
> We acknowledge that we missed the STaR papers and will include them in the related work section. Although we have cited some similar papers [3,4], we appreciate the importance of adding this relevant literature.

---

> ### Author Response · Authors · 2024-08-06
> **Rebuttal Supplement: Templates, Examples and References**
>
> Due to the word limit, I am providing the prompts, examples, and references used in our rebuttal for reviewer SX7Z in this comment.
>
> ## A. Rubrics in Prompt Templates
> As mentioned in our response to **[W5]**, the prompt templates for PRM and ORM are detailed in Appendix A.5 , with specific rubrics for PRM and ORM are
>
> - PRM:
> ```
> You are given a math problem, followed by a step-by-step reasoning process. Your task is to read the problem carefully, understand the solving steps, and check the correctness of the last reasoning step. Output 'True' if the last step is correct, and 'False' otherwise.
> ```
> - ORM:
> ```
> Assess a solution including final answer to a given math problem by following below steps.\n- Evaluate the method used for solving the problem.\n- Review each calculation step for accuracy. Check for computational errors, incorrect formula applications, or arithmetic mistakes.\n- The solution should use all the information provided in the question.\n- Examine the final answer for correctness, considering the calculations and method used.\n.
> ```
>
> ## B. Rollout Example
> Here's a detailed rollout example addressing **[W7]**:
>
> Consider the following GSM-like question:
>
> ```
> Question: Sandy's monthly phone bill expense is equal to ten times her age now. In two years, Sandy will be three times as old as Kim. If Kim is currently x years old, calculate Sandy's monthly phone bill expense.\nIf we know the answer to the above question is 340, what is the value of the unknown variable x?\n
> ```
>
> A node in the second layer could have the following content:
>
> ```
> Answer: We know that Sandy's monthly phone bill is 10 times her age. In two years, Sandy will be 3 times as old as Kim. The sum of Sandy's age now and 2 years is 3 times the sum of Kim's age now and two years.\nSandy's age now is 340/10 = <<340/10=34>>34. In two years, Sandy's age will be 34 + 2 = <<34+2=36>>36.\n
> ```
>
> The parent of this node has the content:
>
> ```
> Answer: We know that Sandy's monthly phone bill is 10 times her age. In two years, Sandy will be 3 times as old as Kim. The sum of Sandy's age now and 2 years is 3 times the sum of Kim's age now and two years.\n
> ```
>
> And one of its fast-rollout paths could be:
>
> ```
> The sum of Sandy's age now and 2 years is 36. The sum of Kim's age now and two years is x + 2.\n36 = 3(x + 2)\n36 = 3x + 6\n3x = 30\nx = 10\n#### 10
> ```
>
> ## C. Reference
> [1]. Zhang, Dan, et al. "ReST-MCTS*: LLM Self-Training via Process Reward Guided Tree Search." arXiv preprint arXiv:2406.03816 (2024).
> [2]. Yu, Longhui, et al. "Metamath: Bootstrap your own mathematical questions for large language models." arXiv preprint arXiv:2309.12284 (2023).
> [3]. Li, Xian, et al. "Self-alignment with instruction backtranslation." arXiv preprint arXiv:2308.06259 (2023).
> [4]. Guo, Hongyi, et al. "Human-instruction-free llm self-alignment with limited samples." arXiv preprint arXiv:2401.06785 (2024).

---

> > ### Comment · Reviewer_SX7Z · 2024-08-09
> > **Thank you for your comments.**
> >
> > I thank the authors for these comments and apologise for not seeing the training section earlier.
> >
> > My concerns about the writing and lack of correct ablations have been re-enforced by the other reviewers, I believe stand.
> >
> > I still think your baselines are wrong, you want to show your method relative to other methods either generates more informative datapoints (similar to compare to Lightman's Process-based supervision) or that this is just a method for boostrapping from a weak learner (in which case i imagine using BoN with these new value functions should get a similar result).
> >
> > The paper is still of weak quality in my opinion but the results are truly exciting. I think if the authors can make a deliberate effort to tidy up writing and compare to baselines with their base models.
> >
> > I'm updating my score to an accept because the result is really good but this paper still really lacks clear communication or generating knowledge on whats happening here.

---

> > > ### Author Response · Authors · 2024-08-10
> > > **Thank you for your feedback**
> > >
> > > Thank you for your constructive suggestions! We truly appreciate your feedback and will work on improving the writing to make the paper clearer and more accessible.
> > >
> > > Regarding the ablations, could you please specify any particular ablations you feel may be missing? In addition to the ablations we included in our submission (with and without certain components), we have also provided more detailed ablations, including hyperparameter controls, in our response to Reviewer PPNk. We would be happy to consider incorporating any additional specific ablations you might suggest.
> > >
> > > As for the baselines, we have included a similar discussion in Section 5.3 and Figure 2. The result demonstrates that, in self-improving, using \etaMCTS for data collection outperforms Reranking (BoN) in various aspects, including the improved policy itself, as well as when combined with reranking and \etaMCTS in both iterations 1 and 2. We believe this evidence shows that our method generates more informative data points.
> > >
> > > Thank you again for your valuable feedback. We will keep refining our paper to enhance its clarity and overall presentation.

---

### Decision · Program_Chairs · 2024-09-25

**Decision:**

Accept (poster)

**Comment:**

The paper draws from AlphaGo to introduce AlphaLLM, a combination of search, simulation, and critic for the self-improvements of LLMs. AlphaLLM uses Monte Carlo Tree Search (MCTS) in a self-improving loop to improve the performance of LLMs. Results on MATHS and GSM8K seem promising. In spite of the interesting results, the reviewers have been concerned with the clarity of the paper, and the presentation of methods, details of how the approach works in the main paper. We recommend the paper for acceptance as a poster conditional on the implementation of the following:
- bring appendix 6 to the main paper
- include responses to Reviewer PPNk in the main paper, including the relevant citations and the clarificaitons
- all reviewers requested the authors include more implementation details, evaluation, and the code
Thank you